# Development of Highly Sensitive and Thermostable Microelectromechanical System Pressure Sensor Based on Array-Type Aluminum–Silicon Hybrid Structures

**DOI:** 10.3390/mi15091065

**Published:** 2024-08-23

**Authors:** Min Li, Yang Xiao, Jiahong Zhang, Qingquan Liu, Xianglong Jiang, Wenhao Hua

**Affiliations:** 1School of Integrated Circuits, Nanjing University of Information Science and Technology, Nanjing 210044, China; limin_nuist@nuist.edu.cn (M.L.); qqliu@nuist.edu.cn (Q.L.); huawenhao777@163.com (W.H.); 2Jiangsu Collaborative Innovation Center on Atmospheric Environment and Equipment Technology, Nanjing University of Information Science and Technology, Nanjing 210044, China

**Keywords:** MEMS, sensor array, aluminum–silicon hybrid structures

## Abstract

In order to meet the better performance requirements of pressure detection, a microelectromechanical system (MEMS) piezoresistive pressure sensor utilizing an array-type aluminum–silicon hybrid structure with high sensitivity and low temperature drift is designed, fabricated, and characterized. Each element of the 3 × 3 sensor array has one stress-sensitive aluminum–silicon hybrid structure on the strain membrane for measuring pressure and another temperature-dependent structure outside the strain membrane for measuring temperature and temperature drift compensation. Finite-element numerical simulation has been adopted to verify that the array-type pressure sensor has an enhanced piezoresistive effect and high sensitivity, and then this sensor is fabricated based on the standard MEMS process. In order to further reduce the temperature drift, a thermodynamic control system whose heating feedback temperature is measured by the temperature-dependent structure is adopted to keep the working temperature of the sensor constant by using the PID algorithm. The experiment test results show that the average sensitivity of the proposed sensor after temperature compensation reaches 0.25 mV/ (V kPa) in the range of 0–370 kPa, the average nonlinear error is about 1.7%, and the thermal sensitivity drift coefficient (TCS) is reduced to 0.0152%FS/°C when the ambient temperature ranges from −20 °C to 50 °C. The research results may provide a useful reference for the development of a high-performance MEMS array-type pressure sensor.

## 1. Introduction

A microelectromechanical system (MEMS) piezoresistive pressure sensor is a miniature and low-power interface of a forward channel that has many engineering applications, especially in the fields of biomedical, industrial control, meteorological detection, and internet of vehicles [1,2,3,4,5,6,7]. However, MEMS piezoresistive pressure sensors made of traditional piezoresistive materials, such as bulk silicon, generally have inherent disadvantages, such as low accuracy, high temperature drift, and low sensitivity. For example, the sensitivity of a piezoresistive sensor with polysilicon thin film designed by Kumar et al. can reach only 0.011 mV/(V·kPa) in the range of 0–30 Bar [8]. In order to further improve the sensitivity of the pressure sensor, researchers have tried to introduce silicon nanowires and graphene and carbon nanotubes into the pressure sensor as piezoresistive materials [9,10,11,12]. Zhang et al. designed a new diaphragm structure by using SiNW as a piezoresistor, which increased the sensitivity of the device in the pressure range of 0–120 mmHg by 2.5 times [13]. However, due to the high production cost, complex production process, and low stability of the sensor made of the above materials, it is difficult for it to be widely used in mass production [14,15,16]. Later, Basov demonstrated another version of pressure sensor chip structures that can simultaneously significantly increase sensitivity by changing the electrical circuit using BJT [17] and reduce temperature characteristics by using MOSFETs in bridge circuits instead of piezoresistors [18]. It is worth noting that Rowe et al. innovatively combined metal and semiconductor to design a novel hybrid structure, which has a greater piezoresistive effect than doped silicon [19]. Afterwards, Ngo developed a MEMS pressure sensor with a metal–silicon hybrid structure as the sensing element, which proved that the hybrid structure has a higher piezoresistive coefficient than traditional bulk silicon [20]. Recently, the sensitivity of an aluminum–silicon hybrid structure pressure sensor designed by Xie can reach 0.283 mV/(V·kPa) [21]. This significantly enhanced sensitivity is attributed to the reduction in the width of the silicon piezoresistive strip. Nevertheless, this also causes the resistance of the pressure-sensitive element to be too small, which affects the stability of the sensor. In order to solve this problem, a long-strip aluminum–silicon hybrid structure may be introduced to increase the resistance while maintaining sensitivity. Unfortunately, a new problem may appear: the size and nonlinear error of this kind of sensor chip will be significantly increased, and the measurement range will be reduced. It is worth noting that the array-type arrangement of pressure-sensing elements is also a common solution to improve the performance of different kinds of sensors, especially piezoresistive sensors [22,23,24,25], which can be applied to the MEMS pressure sensor chip utilizing aluminum–silicon hybrid structures. Firstly, the array-type arrangement can reduce the chip volume compared with the long-strip aluminum–silicon hybrid structure on the strained membrane. Zhao et al. designed a microfabricated shear sensor array with four-by-four groups for pressure gradient calibration, which can achieve a small chip size of 10 mm × 10 mm [26]. On the other hand, the design of a single strained membrane for one element of an aluminum–silicon hybrid structure array can enhance the performance of linearity, while the strained membrane with a larger area may cause a more serious nonlinear error [27]. Finally, the equivalent stress, which is related to sensitivity, ought to attenuate towards the sides of the strained membrane, so the high concentrated stress area (HCSA) must appear in the middle area along the edge of the strained membrane [28]. The attenuation of equivalent stress on the stress-sensitive long-strip aluminum–silicon hybrid structure can only occur on a membrane with a larger area. The design of array-type aluminum–silicon hybrid structures with smaller strained membranes can keep stress uniform on every element of the sensor array, so the attenuation of sensitivity can be eliminated. There are a large number of different types of mechanical stress concentrator designs for complex-profile membranes, which allows one to find an improved balance between high sensitivity and low nonlinearity error [29,30,31,32]. 

In addition, in view of the unavoidable temperature drift problem of the aluminum–silicon hybrid structure, software algorithms and hardware compensations are usually used in most research work. For example, Yao et al. proposed a temperature compensation technology based on passive resistance that uses differential equations to calculate compensation parameters. The experimental results show that this compensation technique with small temperature drift is hopefully suitable for measurement environments with large temperature changes [33]. The hardware compensation has good dynamic characteristics and strong real-time performance, but unfortunately, the design of the processing circuit is complicated, and the flexibility is poor. Therefore, it is difficult to put into commercial production. In addition, Zhou et al. combined a BP neural network and an RBF neural network to propose a new sensor temperature compensation system. Their experimental data show that the measurement accuracy of the sensor improved from 0.7% FS to 0.2% FS [34]. As far as software compensation is concerned, this method has many advantages, such as low temperature, convenient transplantation, and low cost. However, software compensation requires a large amount of experimental data for training and learning, and when the sensor characteristics change, it needs to be re-modeled. Later, the limitations of software and hardware compensation made researchers turn their attention to the constant measurement environment. Du et al. designed a constant temperature control system based on a cavity structure. The pressure sensor is located inside the closed cavity, and the underside of the sensor is controlled by a heating plate. Experiments show that the absolute value of the measurement error is within 0.2 hPa at different external ambient temperatures from −45 °C to 45 °C [35]. Certainly, with the development of the micromachined process, the device design of the temperature-compensated structure is a feasible solution for temperature drift compensation. Seo et al. also realized the temperature compensation calibration of a piezoelectric pressure sensor by manufacturing a diaphragm platinum thermistor [36]. In addition, Mohammed et al. designed an n-type piezoresistive three-dimensional stress sensor with full temperature compensation. Since the full-circular varistor is close to zero stress on the silicon plane, local temperature changes in the chip can be detected separately for compensation [37].

In this paper, an array-type pressure sensor chip using aluminum–silicon hybrid structures is developed. Compared with traditional bulk silicon, the aluminum–silicon hybrid structure shows a better piezoresistive effect. This chip integrates a 3 × 3 array-type strained membrane. The aluminum–silicon hybrid structure array on the film and that outside the film are connected in series, which is equivalent to a lengthened aluminum–silicon hybrid structure. The array-type arrangement reduces the size of the chip and manufacturing cost while avoiding the attenuation of sensitivity and resistance. In order to eliminate the temperature drift of the sensor, this paper designs a temperature reference structure array and an external constant temperature control system based on a proportional–integral–derivative (PID) controller. Firstly, the aluminum–silicon hybrid structure on the strained membrane is employed as a pressure measurement structure, and the aluminum–silicon hybrid structure outside the strained membrane is used as a symmetrical temperature compensation structure. These two structures can cancel each other out in temperature drift. Secondly, due to the temperature-sensitive nature of the temperature compensation structure, it can replace a general temperature sensor for temperature measurement when the environment temperature is compensated for. The measurement result is handed over to the main controller for the PID calculation, and then the heating plate is controlled to achieve external constant temperature control.

## 2. Principle of Hybrid Structure

The aluminum–silicon hybrid structure is a metal–semiconductor hybrid structure, and its working principle and measurement method are different from those of traditional silicon piezoresistive sensors. Figure 1 shows a schematic diagram of the aluminum–silicon hybrid structure. It mainly includes a silicon piezoresistive strip and an aluminum shunt, which are conjoint with ohmic contact. After uniaxial stretching is applied, the tensile strains along the width of the hybrid structure can deflect the current away from the aluminum shunt on account of this stress-induced anisotropy in the silicon conductivity [19,20,21]. From the perspective of macroscopic performance, the resistance is significantly increased, which can be used to achieve pressure measurement. This is an overall effect, and the piezoresistance enhancement has nothing to do with the change in the contact area between silicon and aluminum due to stress. 

It can be seen from Figure 1 that the aluminum–silicon hybrid structure is formed by contacting a doped silicon strip whose length, width, and thickness are 2 L, W, and h with an aluminum shunt with a length of 2 L. The overall resistance is called the hybrid transfer resistance *R_Z_*. The current *I*_0_ flows into the doped silicon strip along the X-axis direction, and *l* is the distance between the two points of the measured voltage. The stress σ acts on the aluminum–silicon hybrid structure along the Y-axis. Considering that the stress in the shear direction is very small, and the transverse and longitudinal piezoresistive coefficients are approximately equal, that is, $\Pi_{l}\approx -\Pi_{t}$, through the weighting function of the mixed resistivity, the mixed transfer resistance of the aluminum–silicon hybrid structure *R_Z_* can be calculated as [38]: (1)$$R_{Z}=\frac{\rho_{0}\sqrt{1-\prod^{2}\sigma }}{h}\cdot \frac{sin\;h\left(\sqrt{\frac{1+\prod \sigma }{1-\prod \sigma }}\cdot \frac{\pi l}{2W}\right)}{cos\;h\left(\sqrt{\frac{1+\prod \sigma }{1-\prod \sigma }}\cdot \frac{\pi L}{2W}\right)},$$
where *ρ*_0_ represents the resistivity calculation formula of the silicon strip without stress, ∏ is the piezoresistance coefficient after simplified calculation, and *σ* is the stress. It can be seen that the mixed transfer resistance of the aluminum–silicon hybrid structure is related to the stress. According to the sensitivity formula, the stress sensitivity *K_Z_* of the aluminum–silicon structure can be calculated as:(2)$$K_{Z}=\frac{1}{R_{Z}}\frac{\partial R_{Z}}{\partial \sigma }=\prod \cdot S_{G},$$

In Formula (2), *S_G_* is the geometric magnification factor, and its calculation formula is:(3)$$S_{G}=\frac{\pi L}{2W}\tan h\frac{\pi L}{2W}-\frac{\pi l}{2W}\cot h\frac{\pi l}{2W},$$

It can be seen from Formula (3) that *S_G_* is proportional to π(L−l)/2W. It can be seen from Formula (2) that the stress sensitivity of the aluminum–silicon hybrid structure is the product of the piezoresistive coefficient and the geometric amplification factor, so it has an enhanced piezoresistive effect compared to traditional silicon.

Different from the traditional bridge measurement method, the aluminum–silicon hybrid structure is powered by a constant current source, and the pressure change information is obtained by measuring the inner pin voltage, as shown in Figure 1. The constant current *I*_0_ enters the aluminum–silicon hybrid structure from the pin Iin and flows out from the Iout pin. The *U*_+_ and *U*_−_ pins are used to measure the voltage difference. Assuming that the initial output voltage when there is no external force is *U*_0_, and under the action of external force *P*, the output voltage of the aluminum–silicon hybrid structure is *U_P_*, then the sensitivity *S* of the aluminum–silicon hybrid structure can be defined as:(4)$$S=\frac{U_{P}-U_{0}}{U_{0}\cdot P}=\frac{I_{P}\cdot R_{Z}(P)-I_{0}\cdot R_{Z}(0)}{I_{0}\cdot R_{Z}(0)\cdot P},$$
Among them, *I*_0_ and *I_P_* are the output of the constant current source before and after the pressure. *R_Z_* (0) and *R_Z_* (*P*) are the transfer resistances of the aluminum–silicon hybrid structure before and after applying the pressure, respectively. Ideally, the current output of the constant current source is stable, that is, *I_P_* = *I*_0_, so the voltage change can be converted into the change in the hybrid transfer resistance and then into the pressure change.

## 3. Realization of Array-Type Sensor

### 3.1. Structure Design of the Array-Type Sensor

Figure 2 gives the structure diagram of the MEMS array-type pressure sensor chip based on the aluminum–silicon hybrid structures. Figure 2a illustrates its cross-sectional view. Each element of the array-type sensor has one stress-sensitive aluminum–silicon hybrid structure on the strain membrane for measuring pressure and another temperature-dependent structure outside the strain membrane for measuring temperature and temperature drift compensation. In order to increase the effective length and resistance value, a 3 × 3 array layout is applied to reduce the chip size and achieve a longer aluminum–silicon hybrid structure. As displayed in Figure 2b, the array-type sensor is composed of nine identical aluminum–silicon hybrid structure units connected in series through wires. This array is divided into two parts: a measurement structure array and a compensation structure array. When a 1 mA constant current source is applied to the chip, the output measurement of the array-type sensor is realized through the pins $U_{M+}$, $U_{M-}$, $U_{REF+}$, and $U_{REF-}$. In this case, the sensitivity of the sensor can be improved by integrating multiple sets of sensing units. Figure 2c displays an aluminum–silicon hybrid structure unit. Each unit includes an aluminum–silicon hybrid structure for pressure measurement, an aluminum–silicon hybrid structure for temperature compensation, and a strained membrane. Each aluminum–silicon hybrid structure has four pin terminals. The two outermost terminals are connected to a constant current source, and the inner two terminals can be used as differential pressure measurement ports. Similar to piezoresistive sensors made of bulk silicon, the aluminum–silicon hybrid structures also have temperature drift. The equivalent series resistance (*ESR*(*T*)) of the hybrid structures obtained by measuring the voltage between the outermost inner needles can be expressed as:(5)$$ESR(T)=\rho_{0}(T)\cdot \frac{\sqrt{\left(1+\Pi_{l}(T)\sigma )\cdot (1+\Pi_{t}(T)\sigma \right)}}{h}\cdot \frac{\sin h\left(\sqrt{\frac{1+\Pi_{t}\left(T\right)\sigma }{1+\Pi_{l}(T)\sigma }}\cdot \frac{\pi l}{2W}\right)}{\cos h\left(\sqrt{\frac{1+\Pi_{t}\left(T\right)\sigma }{1+\Pi_{l}(T)\sigma }}\cdot \frac{\pi L}{2W}\right)},$$

It can be seen from Figure 2 that the strained film is located in the center of each sensing unit, and the aluminum–silicon hybrid structure used for measurement is located at the edge of the strained film. The aluminum–silicon hybrid structure for temperature compensation is outside the strained membrane. When the doping concentration and manufacturing process are consistent, the resistance values of the two equivalent aluminum–silicon hybrid structures are equal. When the temperature is constant, there is no external pressure, and the constant current source is used for the power supply, the output is *U*_0_. When the temperature changes, temperature drift changes of the two equivalent aluminum–silicon hybrid structures are both $\Delta U_{T}$, and the output voltage is $U_{\mathrm{out}}=U_{0}+\Delta U_{T}$. When an external pressure *P* acts on the strained film, the output voltage of the equivalent hybrid structure used for measurement is $U_{M}=U_{0}+\Delta U_{T}+\Delta U_{P}$, and the output voltage of the equivalent hybrid structure for temperature compensation is still $U_{REF}=U_{0}+\Delta U_{T}$. The difference U between $U_{M}$ and $U_{REF}$ is $\Delta U_{P}$. In this way, temperature-induced output change $\Delta U_{T}$ can theoretically be eliminated.

When the side length and thickness of the membrane are the same, the square membrane can withstand greater stress, and the sensitivity of the produced sensor is higher than that of the circular membrane. In order to obtain good linearity, the central deflection *ω_max_* of the strained membrane and the maximum stress *σ_max_* of the membrane can be obtained by the theory of small deflection as:(6)$$\omega_{max}=0.0151\frac{Pa^{4}}{Eh^{3}}\left(1-\mu^{2}\right),$$
(7)$$\sigma_{max}=0.3078\frac{Pa^{2}}{h^{2}},$$
Here, *E* is the Young’s modulus, *h* is the film thickness, *µ* is the Poisson’s ratio, a is the side length of the square film, and *P* is pressure applied to the film. For a square film, the side length and thickness meet [7]:(8)$$h\geq {\left(\frac{0.015\times \left(1-\mu^{2}\right)P}{0.3E}\right)}^{\frac{1}{4}}\times a,$$

In Formula (8), *μ* = 0.278, *E* = 170 GPa, and *P* = 1000 KPa. Here, the thickness of the membrane we chose is 20 μm. Through calculation, the value range of the side length a of the film is *a* ≤ 1182 μm. Based on this, the designed single strained film has a size of 900 μm × 900 μm and a thickness of 20 μm.

The sensor silicon cup designed in this paper is composed of nine identical C-type silicon cups, and each aluminum–silicon hybrid structure unit is placed on the top of the silicon cup. The following is a theoretical design of the window of a single silicon cup. As shown in Figure 2a, *H* is the thickness of the bottom silicon and the middle oxide layer; *b* is the size of the silicon cup window; and *θ* = 54.74° is the corrosion angle. According to the geometric relationship in Figure 2a, the C-type silicon cup satisfies the relationship formula:(9)$$H-h=\frac{1}{2}(b-a)\tan\theta$$

Substituting the parameters of the silicon cup, the window size b of the silicon cup can be calculated to be 1792 μm. Based on the above theoretical calculations, the basic parameters of the array-type sensor can be determined as follows: The silicon substrate of the sensor is a nine-square silicon cup window with a size of 1792 μm × 1792 μm. The height of the silicon cup is 650 μm. The top layer area of the sensor is 9135μm × 10,858 μm. Nine aluminum–silicon hybrid units of 1828 μm × 1828 μm are placed on the surface. The strain film inside the cell is 900 μm × 900 μm, and the thickness is 20 μm.

### 3.2. Finite Element Analysis

In order to verify the feasibility of the scheme, ANSYS finite element simulation software (version 14.5, ANSYS Inc., Canonsburg, PA, USA) was employed to build the finite element structure model and simulate the stress distribution of the array-type sensor using aluminum–silicon hybrid structure units. The unstructured tetrahedron is used for mesh division, with denser mesh division in the area near the aluminum–silicon hybrid structure and lower density division in other areas. This can improve simulation accuracy and shorten finite element simulation time. Figure 3 is the displacement cloud diagram and the stress distribution cloud diagram of the aluminum–silicon hybrid structure array under a pressure of 100 kPa in the direction perpendicular to the X-Y plane downward. It can be observed that the displacement and stress of the temperature compensation structure are almost zero, indicating that they are not affected by external pressure.

In order to calculate and analyze the sensitivity of the array-type sensor, we change the pressure to obtain the stress distribution of the aluminum–silicon hybrid structure unit. Then, we derive the stress and pressure data and draw the relationship curve between internal stress and external pressure, as shown in Figure 4a. It can be seen from the figure that when the external pressure increases in the range of 0~370 kPa, the internal stress of the measurement structure increases linearly and reaches the MPa level. However, almost no stress is generated inside the temperature compensation structure. Substituting the obtained stress data into Equation (1), the hybrid transfer resistance of the aluminum–silicon hybrid structure array corresponding to the stress can be obtained. The calculation parameters include the following: the stress-free resistivity is 0.0435 Ω∙cm, the piezoresistance coefficient is 66.5 × 10^−11^ Pa^−1^, the effective silicon thickness is 1 μm, and the width is 28 μm. In addition, the L and l of the equivalent hybrid structure of the measurement array are 1350 μm and 1260 μm, respectively. After calculation, the relationship between the transfer resistance of the array and the external pressure is obtained, as shown in Figure 4a. The initial resistance of the array-type pressure sensor designed in this paper is 26.4 Ω. As the pressure continues to increase, its transfer resistance shows a linear increasing trend. According to the sensitivity Formula (4), the theoretical sensitivity of the array-type sensor can be calculated to be about 0.318 mV/(V·KPa).

### 3.3. Process Flow and Packing Design

The array-type sensor utilizing aluminum–silicon hybrid structures was fabricated by using a standard MEMS process. A circular p-type <100> SOI wafer with a radius of 6 inches was selected as the raw material; the thickness of its silicon device layer, intermediate buried oxide layer, and substrate layer are individually 5 µm, 1 µm, and 650 µm, respectively. Figure 5 briefly exhibits the fabrication process of the designed sensor chip. The main process consists of the following steps:

Step(a): wafer preprocessing. The SOI wafer was placed in 1NH_4_OH: 1H_2_O_2_: 5H_2_O solutions for removing particle impurities and in 1HCl: 1H_2_O_2_: 6H_2_O solutions for removing inorganic contamination, respectively, and repeatedly cleaned with deionized water. Then, the cleaned wafer was placed in diluted HF solution to remove the natural oxidation of its surface.

Step(b): ion implantation and annealing. The 10^18^ cm^−3^ Boron ions were injected into the device layer at an angle of 10° and 20 KeV energy. Then, the SOI wafer was placed in a high-temperature annealing furnace at 900 °C for 30 min to ensure that the boron ions were uniformly distributed.

Step(c): thermal oxidation. The wafer was placed in a heating furnace and thermally oxidized at a temperature of 1200 °C. The oxidation thickness is about 1 µm, which can be used as a passivation layer for the device layer.

Step(d): lithography and etching. The photoresist was spin-coated, soft-baked, exposed using the mask, and developed using standard photolithography. Afterwards, the silicon dioxide passivation layer and the silicon device layer were dry-etched by an inductively coupled plasma (ICP) process in a mixed gas of SF_6_/N_2_. Then, the silicon piezoresistive strip and its electrode terminal (lead-out pins) were formed. 

Step(e): sputtering aluminum. The uniform aluminum layer was prepared by a radiofrequency (RF) magnetron sputtering method and was subsequently patterned to complete metal aluminum strips, pads, and electrodes. The micro ohmic contact with an implantation depth of 1 µm between aluminum and boron-doped silicon was achieved.

Step(f): silicon cup formation. After lithography and etching of the substrate silicon dioxide, a silicon cup window was formed on the back side, and then tetramethylammonium hydroxide (TMAH) by addition of parts of 3% ammonium persulfate was employed as an etching solution for the silicon cup. The top of the silicon cup is the strained membrane, and the size is 900 μm × 900 μm × 20 μm.

Step(g): oxygen removal. The silicon dioxide etching solution was prepared using hydrofluoric acid (50%) and ammonium fluoride (40%) in a weight ratio of 1:7 to remove the underlying silicon dioxide at the bottom of the silicon cup.

Step(h): anodic bonding. The wafer was bonded to a 300 μm thick borosilicate glass substrate (Pyrex 7740) by using anodic bonding technology under the vacuum conditions of 400 °C temperature and 1000 V forward voltage. 

Finally, a separate array-type sensor chip was cut from the wafer by using a dicing saw. The physical diagram of the sensor chip is presented in Figure 6. It was pressure-welded and packaged on the printed circuit board (PCB) by using 502 glue for calibration and performance evaluation, as shown in Figure 6a,b. The back of one unit of the chip under the microscope is illustrated in Figure 6c.

## 4. Experiment Results and Discussion

### 4.1. Platform Construction

In order to measure the static characteristics of the pressure sensor chip, an experimental measurement platform was built. The experimental instrument was composed of a Fluke PPC-4 air pressure generator (Fluke Corporation, Everett Reed, WA, USA), a const162 desktop air pump (ConST Instruments Technology Inc., Beijing, China), an Agilent 34410A digital multimeter (Agilent Technologies, Inc., Santa Clara, CA, USA), a high- and low-temperature Test box (Cliphyco Instruments Co., Limited, Hong Kong, China), and an upper computer (Lenovo Group Limited, Beijing, China). The physical map is shown in Figure 7a. The temperature hardware compensation structure only compensates for the chip itself. We also need to compensate for the ambient temperature. With this in mind, our approach is to feed back the temperature signal measured by the temperature compensation array to the microcontroller unit (MCU) and then control the heating system to work through the PID algorithm so that the array sensor works at a stable ambient temperature. The heating power device is a polyimide heating diaphragm. The schematic diagram of the thermodynamic control system is displayed in Figure 7b.

### 4.2. Test Results

Figure 8a is the theoretical value of the temperature compensation structure obtained by Formula (5) and the actual measured voltage output value at different temperatures. The measured results are consistent with the negative temperature coefficient characteristics of the aluminum–silicon hybrid structure [21]. Thus, the current chip temperature can be calculated by measuring the voltage value across the temperature compensation array to perform temperature compensation. This method can replace the traditional temperature sensor, which can not only reduce the cost but also reduce the temperature measurement error caused by the distance. In order to study the influence of external temperature and pressure on the output characteristics of the sensor, we consider the sealing ability of the gel encapsulation, the pressure range of 0~370 kPa, and the temperature range of −20 °C to 50 °C are selected for testing and evaluation. Figure 8b shows the voltage output curved surface of the pressure-measuring structure when the pressure and temperature change. The output voltage of each pressure sampling point gradually decreases with the increase in temperature, indicating that the array-type pressure sensor made of aluminum–silicon hybrid material has temperature drift.

As plotted in Figure 8c, during the temperature rise from −20 °C to 50 °C, the sensitivity of the pressure sensor showed an overall upward trend, with the highest rising to 0.252 mV/(V·KPa). Among them, the maximum sensitivity change rate from −20 °C to 50 °C is 2.7%. The nonlinear error of the sensor has no obvious change rule, which is in the range of 1.5~2.3%. In order to study the temperature drift characteristics of the sensor, we used the 20 °C output curve as a reference and the 10 kPa measurement point as the zero-input point and calculated the thermal zero drift coefficient and thermal sensitivity drift coefficient at each temperature, as shown in Figure 8d. The thermal zero drift coefficient of the sensor has a relatively stable curve with temperature, which is basically maintained at about 4 × 10^−1^ %FS/°C. In contrast, the thermal sensitivity drift coefficient curve has no obvious law, and the overall distribution is axisymmetrical at 20 °C.

Figure 9a shows the change in the output of the sensor with pressure application and release. When there is no external pressure, the sensor has no large fluctuations. When there is an external pressure, the output voltage increases significantly and reaches a stable state after 5~6 s. After canceling the external pressure, the output voltage begins to drop and stabilizes in the initial state after 2~3 s. Through repeated tests on the output under different external pressures, it can be seen that the array-type pressure sensor designed in this paper has a faster response speed and good output stability. Figure 9b gives the relative error value of the sensor’s output voltage under the condition of rising and falling temperatures. The change trend of the five sets of data measured at each node after the temperature rises and falls to stabilize is consistent with the material’s negative temperature coefficient characteristics. The data measured after returning to 20 °C indicate that the output of the array-type sensor has good consistency.

In order to test the reliability of the constant temperature control system, we measured the temperature of the pressure sensor chip under different external temperatures. The specific steps are as follows: we set the target temperature of the constant temperature system to 50 °C, and the external temperature range is −30 °C to 50 °C. The temperature is measured every time the external temperature increases by 10 °C, and the temperature value measured by the temperature compensation array is recorded every 5 s. The temperature value measured by the temperature compensation array is plotted in Figure 10a. When the constant temperature system starts to work, the heating plate starts quickly, and the temperature of the sensor chip rises sharply. Subsequently, as the PID regulates the PWM, the chip temperature oscillates. After a period of time, the chip temperature tends to stabilize. Through testing, it is found that the time it takes to reach a constant temperature of 50 °C is related to the outside temperature. The greater the temperature difference between the external temperature and the target temperature, the longer the required time, and vice versa. Figure 10a also shows the temperature deviation graph from 465 s to 600 s. It shows that the temperature deviations of 30 °C, 40 °C, and 50 °C are the smallest, all within the range of 0.1 °C. The thermostatic effect at other temperatures is a little inferior, but it is also within the range of ±0.3 °C, indicating that the thermostatic control system designed in this paper has high control accuracy.

Figure 10b plots the pressure output curve of the sensor after hardware temperature compensation and constant temperature system compensation. Compared with Figure 8b, it can be seen that this output curve is less affected by temperature, and the sensor output curve at different temperatures has a high degree of coincidence. It shows that the temperature compensation effect is obvious. The calculation shows that the repeatability error after temperature compensation is 2.20%.

The final test results show that the sensitivity of the proposed array-type sensor reaches 0.25 mV/(V·KPa) in the range of 0~370 kPa. Compared with a single aluminum–silicon hybrid structure sensor with a sensitivity of 0.13 mV/(V·KPa) [7], the sensitivity of the aluminum–silicon hybrid structure array-type pressure sensor designed in this paper is about twice that. It can be seen from Figure 10c that the minimum sensitivity of the sensor after temperature compensation is 0.251 mV/(V·KPa), and the maximum is 0.253 mV/(V·KPa). The thermal sensitivity drift coefficient is 1.52 × 10^−2^ %FS/°C in the range of −20 °C to 50 °C, which is less than the thermal sensitivity drift coefficient without compensation. The test data show that the temperature compensation method can effectively suppress the sensitivity drift caused by temperature changes. Figure 10d demonstrates that the nonlinear error after temperature compensation is between 1.29% and 2.09%. The average nonlinear error is 1.7%, which is slightly less than 1.89% without compensation. It can be seen that temperature compensation also improves the linearity of the sensor, to a certain extent. 

## 5. Conclusions

Based on the enhanced piezoresistive effect of an aluminum–silicon hybrid structure, an array-type pressure sensor is proposed. We integrate nine groups of aluminum–silicon hybrid structures on the sensor chip for a smaller sensor size and lower sensitivity attenuation, which contains a stress-sensitive structure and a temperature compensation structure. Moreover, a constant temperature system is designed to suppress temperature drift. The final test results show that the sensitivity of the proposed sensor reaches 0.25 mV/(V·KPa). Compared with a single aluminum–silicon hybrid structure sensor with a sensitivity of 0.13 mV/(V·KPa), the sensitivity of the array-type pressure sensor designed in this paper is about twice that. Certainly, the measured pins of each unit in the 3 × 3 sensor array can be made in the future in order to ensure the functionality of the entire sensor chip when one or several units are invalid in a harsh working environment. Software algorithms can be used to further compensate for temperature drift to improve measurement accuracy and other performance, and the compensation measurement structure and circuit also need to be further optimized in the future.

## Figures and Tables

**Figure 1 micromachines-15-01065-f001:**
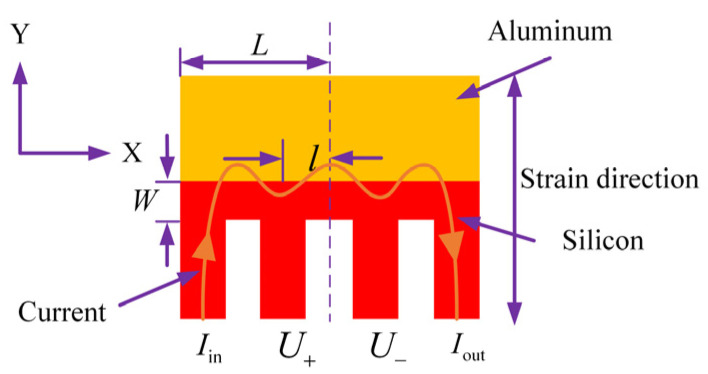
Schematic diagram of the aluminum–silicon hybrid structure.

**Figure 2 micromachines-15-01065-f002:**
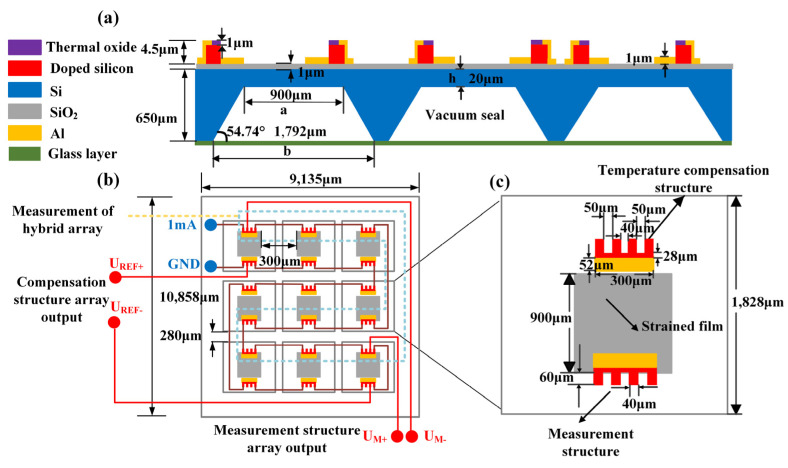
Structure diagram of the array-type aluminum–silicon hybrid structures: (**a**) longitudinal section of the array; (**b**) top view of the array; (**c**) unit structure diagram with dimensions.

**Figure 3 micromachines-15-01065-f003:**
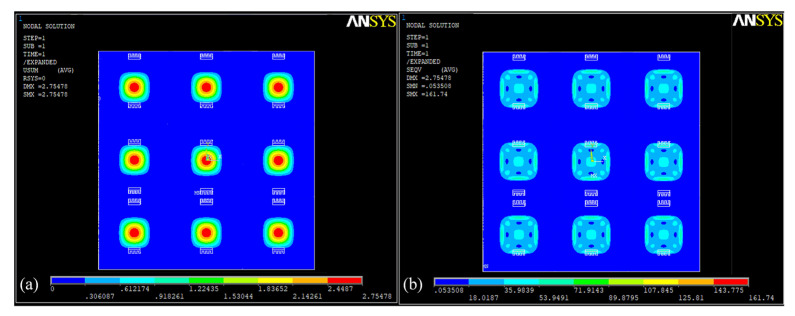
Simulation diagram of aluminum–silicon hybrid array: (**a**) displacement cloud; (**b**) stress distribution cloud.

**Figure 4 micromachines-15-01065-f004:**
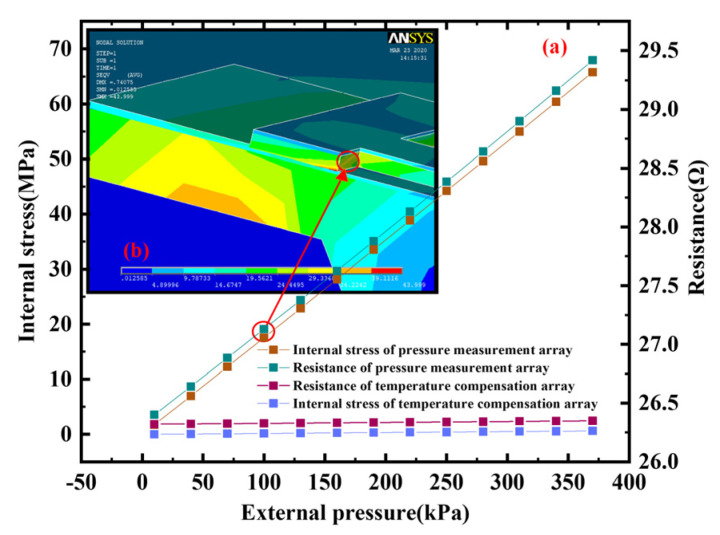
Internal stress and resistance under different external pressures: (**a**) the relationship between the internal stress or resistance of the two structures and the external pressure; (**b**) local stress cloud diagram of the aluminum–silicon hybrid structure at 100 kPa.

**Figure 5 micromachines-15-01065-f005:**
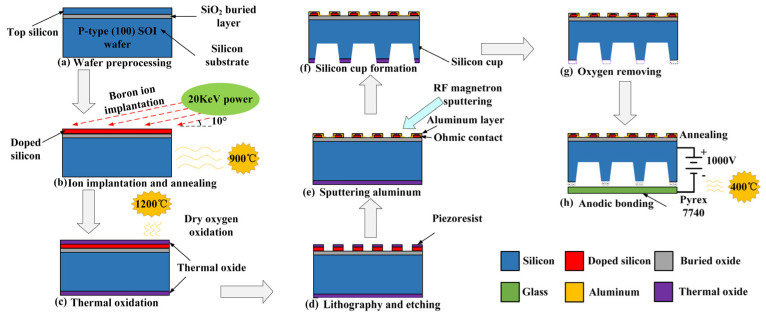
MEMS process flow chart.

**Figure 6 micromachines-15-01065-f006:**
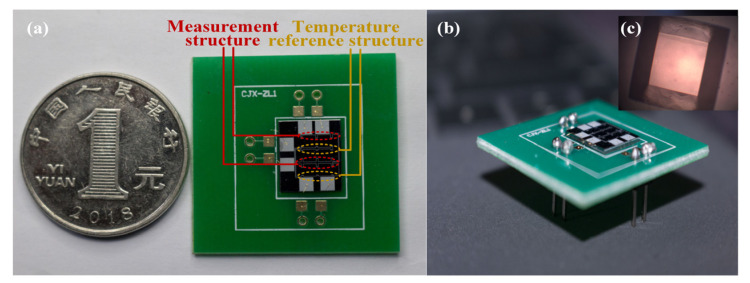
Chip map: (**a**) top view of the array-type sensor chip; (**b**) side view of the array-type sensor chip; (**c**) enlarged view of the back of the chip.

**Figure 7 micromachines-15-01065-f007:**
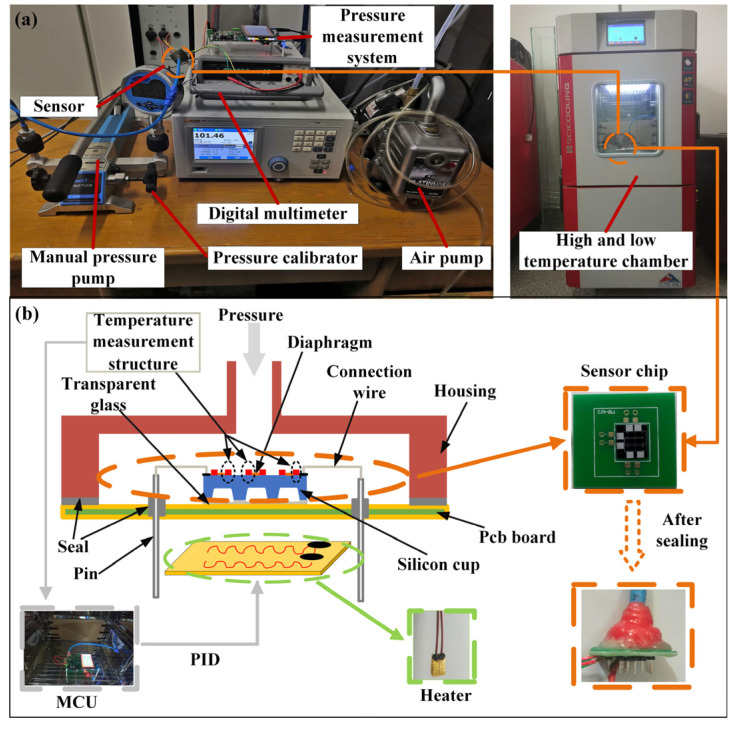
Experimental platform diagram: (**a**) platform physical map; (**b**) schematic diagram of thermodynamic control system.

**Figure 8 micromachines-15-01065-f008:**
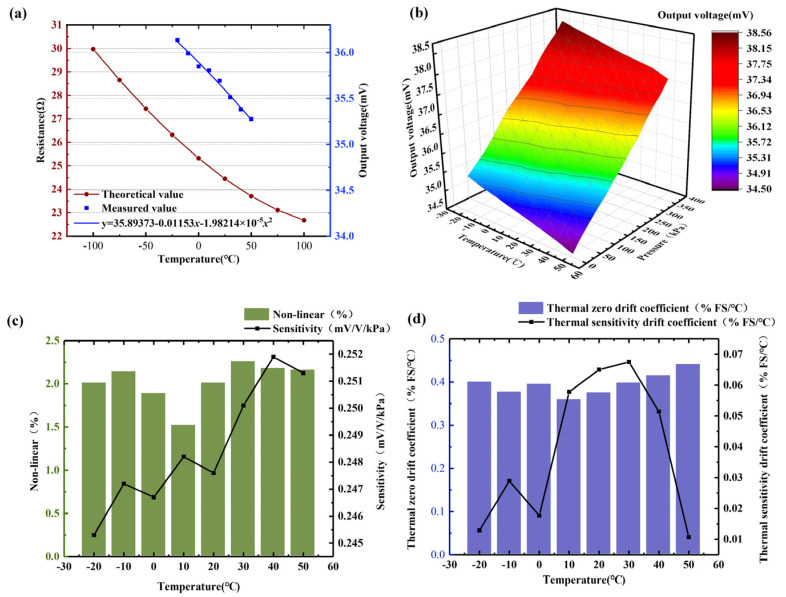
The relationship between sensor output and temperature: (**a**) the theoretical output value and the actual measured value of the temperature compensation structure; (**b**) output value under different temperatures and pressures; (**c**) sensitivity and nonlinear error at different temperatures; (**d**) thermal zero drift coefficient and thermal sensitivity drift coefficient at different temperatures.

**Figure 9 micromachines-15-01065-f009:**
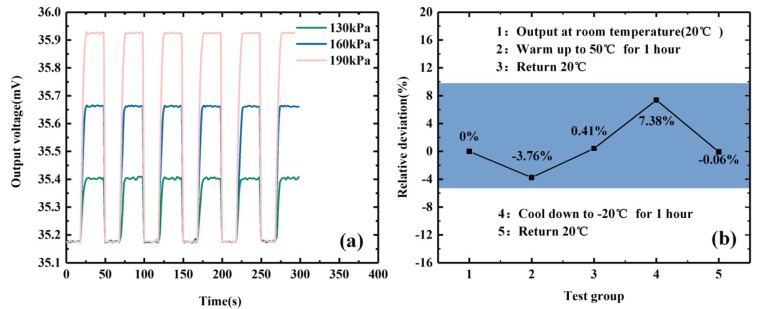
Stability test: (**a**) repeated testing of output under different external pressures; (**b**) relative error test of the output voltage of the sensor under rising and falling temperatures.

**Figure 10 micromachines-15-01065-f010:**
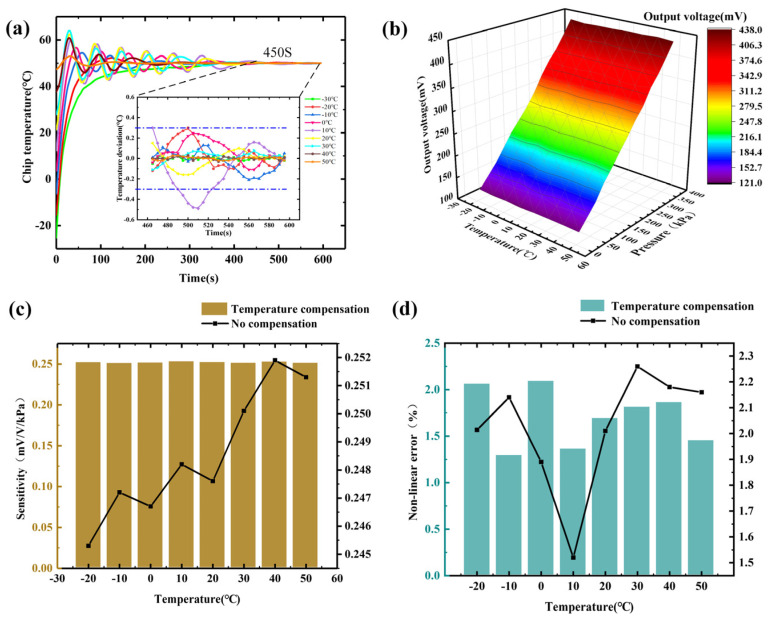
The relationship between sensor output and temperature after temperature compensation: (**a**) temperature change curve with time after temperature compensation; (**b**) pressure output curve after hardware temperature compensation structure and constant temperature system compensation; (**c**) comparison of sensor sensitivity after temperature compensation and sensor sensitivity without temperature compensation; (**d**) comparison of nonlinear error after temperature compensation and nonlinear error without temperature compensation.

## Data Availability

The original contributions presented in the study are included in the article, further inquiries can be directed to the corresponding authors.

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
