# Peer review of "Development of Highly Sensitive and Thermostable Microelectromechanical System Pressure Sensor Based on Array-Type Aluminum–Silicon Hybrid Structures"

_micromachines, 2024, doi:10.3390/mi15091065_

Round 1
Reviewer 1 Report
Comments and Suggestions for Authors
1.The dimensioning in Figure 2 should be checked to meet a certain standard. The "900μm" and "54.74°" in Figure 2 cannot clearly indicate where the size is.
2.The color blocks of the legend for the different materials in Figure 2 (a) should be the same size, and the text should be aligned.
3.Figure 5 is suggested a horizontal arrangement.
4.What does "the 3´3 sensor array" stand for in the conclusion?
Comments on the Quality of English LanguageMinor editing of English language required.
Author Response
Response to Reviewer #1:
We very much appreciate the reviewer for careful reading of our manuscript and valuable suggestions. We have revised the manuscript accordingly. The revised portions were marked in red in the resubmitted manuscript.
Q1: The dimensioning in Figure 2 should be checked to meet a certain standard. The “ 900 µm” and “54.74”in Figure 2 cannot clearly indicate where the size is.
Reply1: Thank you very much! Figure 2 has been further improved in the revised manuscript for better readability (see page 5).
Q2: The color blocks of the legend for the different materials in Figure 2(a) should be the same size, and the text should be aligned.
Reply2: Thank you for your suggestion! Figure 2(a) has been further improved in the revised manuscript for better readability (see page 5).
Q3: Figure 5 is suggested a horizontal arrangement.
Reply3: Thank you very much! Figure 5 has been redrawn by horizontal arrangement in the revised manuscript for better readability (see page 9).
Q4: What does “the 3’3sensor array” stand for in the conclusion?
Reply4: Thank you very much! We regret to this basic error, and it has been modified in the revised version (see page 13).

Reviewer 2 Report
Comments and Suggestions for Authors
The presented paper considers the fabrication and characterization of a highly sensitive and thermostable pressure sensor.
The authors present the array-type aluminum-silicon hybrid structures of the sensor in detail. The design improves the sensitivity and thermal stability of the pressure sensor. Here, the fabrication of the device is very clearly presented. Finally, sensitivity at different temperatures, sensor output after temperature compensation, and stability test results are presented. The experimental results are viable and explicitly explained.
This is a well-written paper with a reasonably good use of the literature. I recommend publication after the following comments are addressed:
1. The unit of sensitivity needs to be unified as mV/(V·KPa).
2. The design parameters are described in the content and graphics dispersedly . It is recommended to use a table to list all the parameters.
3. Can you provide a clear chip diagram in figure 6?
4. The pressure range for this sensitivity of 0.25 mV/(V·KPa) should be decribed clearly in the paper. The description of TCS is also like this.
Author Response
Response to Reviewer #2:
We very much appreciate the reviewer for careful reading of our manuscript and valuable suggestions. We have revised the manuscript accordingly. The revised portions were marked in red in the resubmitted manuscript.
Comments and Suggestions for Authors
The presented paper considers the fabrication and characterization of a highly sensitive and thermostable pressure sensor. The authors present the array-type aluminum-silicon hybrid structures of the sensor in detail. The design improves the sensitivity and thermal stability of the pressure sensor. Here, the fabrication of the device is very clearly presented. Finally, sensitivity at different temperatures, sensor output after temperature compensation, and stability test results are presented. The experimental results are viable and explicitly explained. This is a well-written paper with a reasonably good use of the literature. I recommend publication after the following comments are addressed:
Q1: The unit of sensitivity needs to be unified as mV/(V·kPa).
Reply1: Thank you for your suggestion! The unit of sensitivity has been unified as mV/(V·kPa). (see page 1, 2, and so on)
Q2: The design parameters are described in the content and graphics dispersedly. It is recommended to use a table to list all the parameters.
Reply2: Thank you very much! In the revised manuscript, we have enriched the parameters in Figure 2, and all parameters have been displayed. Due to the large number of parameters, it is easier to understand when combined with the diagram.
Q3: Can you provide a clear chip diagram in figure 6?
Reply3: Thank you for your suggestion! Relevant chip diagram and explanations have been supplemented in the revised version. (see page 9).
Q4: The pressure range for this sensitivity of 0.25 mV/(V·kPa) should be described clearly in the paper. The description of TCS is also like this.
Reply4: Thank you very much! The pressure range for the sensitivity and temperature range for TCS have been described in the revised manuscript. We have added the express of “The experiment test results show that the average sensitivity of the proposed sensor after temperature compensation reaches 0.25 mV/ (V•kPa) in range of 0-370 kPa, the average nonlinear error is about 1.7%, and the thermal sensitivity drift coefficient (TCS) is reduced to 0.0152%FS/℃ during the ambient temperature ranges from -20 ℃ to 50 ℃.” “The final test results show that the sensitivity of the proposed array-type sensor reaches 0.25 mV/(V·KPa) in the range of 0~370kPa. The thermal sensitivity drift coefficient is 1.52×10-2 %FS/℃ in the range of -20 ℃ to 50 ℃, ”(see page 1 and page 10 ).

Reviewer 3 Report
Comments and Suggestions for Authors
Dear Authors,
I am glad that I had the opportunity to familiarize myself with your important new research in the field of implementing MEMS pressure sensors with high sensitivity and low error in the temperature coefficient of the zero signal. It is good and important results for subsequent practical application in production. Unfortunately, for now I have a few questions for you regarding the presentation of the material and a small addition for the evidence base. I will not dwell on specific issues here, all my recommendations are listed in the attached file "micromachines-3047137-peer-review-v1 (Review)".
I wish you success in all your future research!
Reviewers

Author Response
Response to Reviewer #3:
We very much appreciate the reviewer for careful reading of our manuscript and valuable suggestions. We have revised the manuscript accordingly. The revised portions were marked in red in the resubmitted manuscript.
Comments and Suggestions for Authors
I am glad that I had the opportunity to familiarize myself with your important new research in the field of implementing MEMS pressure sensors with high sensitivity and low error in the temperature coefficient of the zero signal. It is good and important results for subsequent practical application in production. Unfortunately, for now I have a few questions for you regarding the presentation of the material and a small addition for the evidence base. I will not dwell on specific issues here; all my recommendations are listed in the attached file "micromachines-3047137-peer-review
-v1 (Review)". I wish you success in all your future research!
Q1: Dear Authors, I am glad that I have the opportunity to familiarize you with interesting and important research in the field of development of highly sensitive and thermostable MEMS pressure sensors. As a researcher directly in this area, I would like to say that you have demonstrated a lot of useful solutions - great job! But it seems to me that some points in the presentation could be corrected or supplemented for a more comprehensive and detailed understanding of the idea of manuscript for future Readers.
Reply1: Thank you very much! The details of the abstract have been enriched. (see page 1).
Q2: Dear Authors, it is very nice to see how you were able to briefly but succinctly describe all the significant trends in the development of silicon MEMS pressure sensors using the piezoresistive method. To slightly expand your review for future Readers, demonstrate another version of pressure sensor chip structures, which can simultaneously significantly increase sensitivity by changing the electrical circuit using BJT [ https://doi.org/10.1109/SENSORS47087.2021.9639504 ], and reduce temperature characteristics by using MOSFETs in bridge circuits instead of piezoresistors [ https://doi.org/10.1002/pssa.201700680 ]. One sentence with the basic parameters will be enough.
Reply2: Thank you very much! To expand our review for future Readers, the relevant literature has been cited in the revised manuscript. “Basov demonstrated another version of pressure sensor chip structures, which can simultaneously significantly increase sensitivity by changing the electrical circuit using BJT [17], and reduce temperature characteristics by using MOSFETs in bridge circuits instead of piezoresistors [18].”(see page 2)
Q3: Dear Authors, I would like to draw your attention to this point, since for example, I, as a Reader, may immediately have simple questions - why does the membrane not have mechanical stress concentrators? There are a large number of different types of mechanical stress concentrator designs for complex-profile membranes, which allow one to find an improved balance between high sensitivity and low nonlinearity error. Here are a few examples [ https://doi.org/10.1049/mnl.2014.0154 , https://doi.org/
10.1088/0957-0233/27/12/124012, https://doi.org/10.1088/1361-6439/ab9581, https://doi.org/10.1109/TIE.2017.2784341] that you should briefly mention in the Introduction for the correctness of the proposed analogues.
Reply3: Thank you for your suggestion! To increase readability, the relevant literature has been cited in the revised manuscript. “There are a large number of different types of mechanical stress concentrator designs for complex-profile membranes, which allow one to find an improved balance between high sensitivity and low nonlinearity error [29-32]. ”(see page 2)
Q4: Dear Authors, I understand that the main emphasis of your research is aimed at reducing the temperature coefficient of the zero signal, but as is known in pressure sensor chips, the more important parameters are uncompensated errors such as temporary stability and temperature hysteresis of the zero signal and sensitivity. Looking ahead a little, I would like to say that I saw your data on temperature hysteresis, but could you expand the information about this important research - 1. Indicate the reasons for the occurrence of hysteresis (after all, the presence of aluminum in close proximity to the thinned part of the membrane is a significant contribution to relaxation from temperature), 2. specify hysteresis for sensitivity.
Reply4: Thank you very much! In future research, we will further test and analyze the relevant results.
Q5: Dear Authors, I apologize for my meticulousness, but when I read such statements, which are given simply as a fact, questions arise: 1. Why exactly these sizes (for each size)? 2. For what pressure range are you considering the chip to operate? 3. What type of pressure: differential, absolute or gauge? 4. What was your goal for errors in nonlinearity and temperature coefficient (drift) - I understand that you will show this later in the model, but tell me what you were aiming for?
Reply5: Thank you very much! The size is an empirical value obtained through theoretical derivation and relevant literature. The pressure range of the chip is related to the size of the diaphragm. We mainly use it for meteorological and tire pressure measurement, with a general working pressure range of 0-400kPa and it is absolute pressure measurement. The purpose of reducing nonlinear errors and temperature drift is to measure pressure as accurately as possible.
Q6: What mesh did you use? Was it even?
Reply6: Thank you very much! The unstructured tetrahedra is used for mesh division, with denser mesh division in the area near the aluminum-silicon hybrid structure and lower density division in other areas. This can improve simulation accuracy and shorten finite element simulation time. The relevant description has been added to the revised manuscript.
Q7: For example, clarify why the plane is (100)? If you use wet etching and don't do diffusion processes to dope the p-n junctions under the piezoresistors (only for connection if I right), then you could take the (110) plane and get vertical walls for etching, thereby reducing the area of the chip. Why (100)?
Reply7: Thank you very much! We chose a (100) crystal surface SOI silicon wafer and used ion implantation doping to form the aluminum-silicon hybrid sensitive structure. This crystal surface has good mechanical properties and thermal chemical stability, which can be beneficial for high-precision pressure measurement.
Q8: This is a very important point in all your research! Packaging can introduce such large impacts that they simply overwhelm all the benefits on the chip. Specify the thickness of the glass you used, as well as the material and method of connection with the subsequent PCB substrate.
Reply8: Thank you very much! The thickness of PYREX 7740 glass plate is 300 μm. The chip is fixed to the PCB with 502 glue. The relevant description has been added to the revised manuscript. (see page 9)
Q9: Dear Authors, another very significant factor before any research (like packaging) is the preliminary operating of the pressure sensor on temperature and pressure. This makes it possible to reduce the mechanical stress of the chip in its subsequent use. Have you done thermo- and barocycling? If yes, please indicate the modes.
Reply9: Thank you very much! Generally speaking, we will place the chip for a long time (such as two years) before conducting measurements, in order to release stress as much as possible. The thermo- and barocycling has also been done. Figure 9 give some results.
Q10: Dear Authors, I have one last rather simple and logical question: how many samples were studied and what is the technological error both in sensitivity and in the studied errors?
Reply10: Thank you very much! There are two sample chips on a silicon wafer with different size and number of arrays,so we didn't compare the relevant errors.

Round 2
Reviewer 3 Report
Comments and Suggestions for Authors
Dear Auhors,
Thank you for your answers. I have got all necessary information so I can recommend to public it in this form.
Kind regards,
Reviewer